# Clinical Advances and Future Directions of Oncolytic Virotherapy for Head and Neck Cancer

**DOI:** 10.3390/cancers15215291

**Published:** 2023-11-04

**Authors:** Zhan Wang, Peng Sun, Zhiyong Li, Shaowen Xiao

**Affiliations:** 1Department of Stomatology, Wenzhou Medical University Renji College, Wenzhou 325000, China; 2School of Basic Medical Sciences, Wenzhou Medical University, Wenzhou 325000, China; sunpeng@wmu.edu.cn (P.S.); lizhiyong02@caas.cn (Z.L.); 3Cixi Biomedical Research Institute, Wenzhou Medical University, Ningbo 315000, China; 4Key Laboratory of Carcinogenesis and Translational Research (Ministry of Education/Beijing), Department of Radiation Oncology, Peking University Cancer Hospital & Institute, Beijing 100142, China

**Keywords:** head and neck cancer, oncolytic viruses, clinical trials, immunotherapy

## Abstract

**Simple Summary:**

Head and neck cancer (HNC) is a significant global health issue, and traditional treatments such as surgery, chemotherapy, and radiation therapy often have limited success, especially in advanced cases. Oncolytic virotherapy (OVT) offers a new approach. Researchers have been working with various viruses, including herpes simplex virus and adenovirus, to target and kill cancer cells while sparing healthy ones. Some viruses have been genetically modified to enhance their tumor-targeting abilities and safety. Clinical trials have shown encouraging results, with improved patient survival rates and minimal side effects. Combining oncolytic viruses (OVs) with other treatments such as chemotherapy or immunotherapy has also demonstrated promise. While challenges such as optimizing dosages and addressing immune responses remain, OVT presents a hopeful avenue for improving HNC treatment in the future.

**Abstract:**

Oncolytic viruses (OVs), without harming normal tissues, selectively infect and replicate within tumor cells, to release immune molecules and tumor antigens, achieving immune-mediated destruction of tumors and making them one of the most promising immunotherapies for cancer. Many clinical studies have demonstrated that OVs can provide clinical benefits for patients with different types of tumors, at various stages, including metastatic and previously untreatable cases. When OVs are used in combination with chemotherapy, radiotherapy, immunotherapy, and other treatments, they can synergistically enhance the therapeutic effects. The concept of oncolytic virotherapy (OVT) was proposed in the early 20th century. With advancements in genetic engineering, genetically modified viruses can further enhance the efficacy of cancer immunotherapy. In recent years, global research on OV treatment of malignant tumors has increased dramatically. This article comprehensively reviews the findings from relevant research and clinical trials, providing an overview of the development of OVT and its application in the clinical treatment of head and neck cancer. The aim is to offer insights for future clinical and fundamental research on OVT.

## 1. Introduction

Head and neck cancer (HNC) refers to malignant tumors that occur within the anatomical region extending from the skull base to above the clavicles and in front of the cervical spine. It ranks as the seventh most common malignancy globally, comprising 5% of all cancers in China [1,2,3]. Among all head and neck malignancies, squamous cell carcinoma (SCC) accounts for approximately 90% [4], primarily originating from the oral cavity, nasal cavity, paranasal sinuses, pharynx, and larynx [3]. Due to the complex anatomical and physiological structures in the head and neck region, head and neck squamous cell carcinoma (HNSCC) exhibits high heterogeneity. More than 60% of patients are diagnosed with advanced-stage disease initially, and after comprehensive treatment, the metastasis or recurrence rate ranges from 40% to 60% [5,6], with a five-year survival rate of less than 50% [7,8,9].

Surgery is the primary treatment modality for HNSCC [3,10,11,12], while radiotherapy and chemotherapy are the main treatment options for inoperable cases, advanced-stage disease, or recurrent cases [13,14,15,16]. Digital techniques, navigation surgery, and artificial intelligence have been integrated into the overall management of HNCs, further enhancing the precision, safety, and effectiveness of treatment plans [17,18,19]. In addition to surgical, radiation, and chemotherapy approaches, targeted therapy, hyperthermia [20], and radioactive particle interstitial brachytherapy [21,22] have been utilized. Immunotherapy [23,24,25] has emerged as a crucial treatment option for HNCs, including immune checkpoint inhibitors (ICIs) [26], antiepidermal growth factor receptor monoclonal antibodies [27], and near-infrared photoimmunotherapy [28,29]. Immunotherapy works by harnessing the patient’s own immune system to activate antitumor immune responses, control and eliminate tumor cells, and reverse tumor immune suppression [30]. Oncolytic virus (OV) immunotherapy involves using viruses to induce tumor cell death, release tumor antigens, and activate the immune system for long-lasting antitumor responses [31,32]. Genetically modified viruses can enhance the effectiveness of oncolytic virotherapy (OVT) through various antitumor mechanisms [33]. This article provides a comprehensive review of the progress in OVT for HNC, discussing its development, therapeutic mechanisms, and prospects.

## 2. Development and Application of OVs

OVs, including various DNA and RNA viruses, can selectively infect tumor cells and replicate within them, leading to their lysis, and do not harm the surrounding normal tissue [30,33,34]. OVs can be broadly categorized into three classes [33,34,35]: (1) natural viruses that can replicate specifically within tumor cells without modification; (2) second-generation OVs achieved through genetic modification, such as deleting viral gene segments, using transcriptional elements as promoters or enhancers, or modifying viral surface proteins; and (3) third-generation OVs created through genetic engineering to express therapeutic genes such as granulocyte-macrophage colony-stimulating factor (GM-CSF). Third-generation OVs integrate advantages to achieve a more extensive oncolytic immune effect [36].

The mechanisms of OV antitumor activity primarily include three aspects [32,33,34,35,36,37,38,39,40,41]: (1) direct virus-mediated cytotoxicity, where the virus specifically infects tumor sites and self-replicates, leading to the infection and destruction of tumor cells without harming normal cells; (2) virus infection that disrupts the tumor vascular system, triggering the influx of neutrophils, causing vascular collapse and tumor cell death; (3) the virus induces chemokines and cytokines, activating local and systemic immune responses, transforming “cold” tumors into “hot” tumors, and inducing immunogenic cell death (ICD). ICD triggered by exposure to OVs results in the release of various molecules, including pathogen-associated molecular pattern molecules (PAMPs), damage-associated molecular pattern molecules (DAMPs), tumor-associated antigens (TAAs), and tumor-associated neoantigens (TANs) [35,38]. The released PAMPs and DAMPs play a crucial role in activating the innate immune response within the tumor microenvironment (TME), contributing significantly to the adjuvant effect on tumor cells. The recognition of PAMPs/DAMPs by pattern recognition receptors (PRRs) in cancer or immune cells initiates the expression of proinflammatory cytokines such as type I interferons (IFNs), interleukin (IL)-1β, IL-6, IL-12, TNF-α, and GM-CSF, and chemokines such as CCL2, CCL3, CCL5, and CXCL10 [31,33]. These chemokines serve to attract neutrophils and macrophages to the sites of infection, while the cytokines activate innate immune cells such as natural killer (NK) cells and dendritic cells (DCs). This, in turn, further stimulates the production of IFNs, TNF-α, IL-12, IL-6, and additional chemokines, thereby amplifying the initial innate response. Consequently, “cold” tumors become “hot” tumors as a result of this immunological transformation. Type I IFNs play a role in increasing the expression of major histocompatibility complex (MHC) class I and II molecules, as well as costimulatory molecules such as CD40, CD80, and CD86 on the surface of DCs. TAAs and TANs released into the environment undergo processing, and are subsequently presented on the surface of antigen-presenting cells (APCs) in association with MHC molecules. The collective action of numerous cytokines and chemokines contributes to the recruitment and activation of antitumor CD8+ T cells and B cells [33,35].

With the continuous development of virology and genetic engineering technology, researchers have been able to genetically edit OVs, significantly improving their safety, specificity, and efficacy in cancer treatment. As a result, four OV drugs have received regulatory approvals for marketing (Table 1). The first OV approved by national regulatory agencies was the unmodified ECHO-7 strain enteric virus RIGVIR [42]. This virus was approved for the treatment of skin melanoma in Latvia in 2004. Oncorine (H101) was approved for the Chinese market in 2005, making it the second OV product worldwide. It is used in combination with 5-fluorouracil and cisplatin to treat nasopharyngeal carcinoma that cannot be surgically removed or has relapsed, becoming the first oncolytic adenovirus (AD) used in clinical practice through intratumoral injection in China [43]. In October 2015, the U.S. Food and Drug Administration approved the genetically modified herpes simplex virus (HSV) product Talimogene laherparepvec (T-VEC), marketed as Imlygic. It became the first OV approved for use in the United States and the European Union, initially for the treatment of advanced melanoma, and later expanded to other malignant tumors, including HNSCC. The United Kingdom national guidelines for the management of head and neck mucosal melanoma recommend that metastatic patients can be treated with chemotherapy or T-VEC [44]. In October 2021, Japan introduced the third-generation OV Delytact (Teserpaturev/G47∆) based on the HSV for the treatment of malignant gliomas.

The history of virus infections leading to tumor regression has ancient roots [45], and contemporary examples continue to emerge. While cancer patients may experience worsened conditions after infection with the SARS-CoV-2 virus [46], some case reports suggest that certain cancer patients, including those with metastatic colon cancer, metastatic renal cell carcinoma, stage III EBV-positive Hodgkin’s lymphoma, NK lymphoma, follicular lymphoma, among others, experienced cancer remission or improvement after SARS-CoV-2 infection [47,48,49,50,51,52]. The antitumor mechanism may be related to the virus-induced autoimmunity, similar to OVT, suggesting that SARS-CoV-2 is a potential OV [53]. A case report published in 2022 demonstrated spontaneous regression of metastatic salivary gland mucoepidermoid carcinoma in a 61-year-old woman after receiving two doses of the mRNA-1273 vaccine, with a 13% reduction in lung nodules observed on chest computed tomography scans [54].

## 3. Advances of OVT for HNCs

OVT for head and neck tumors is primarily administered through intratumoral or intravenous injection, and has demonstrated good safety profiles in clinical trials. Currently, viruses used in clinical trials for HNCs include DNA viruses such as AD, HSV, and vaccinia virus (VV), as well as RNA viruses such as reovirus (RV), vesicular stomatitis virus (VSV), and measles virus (MV) [37,44,55] (Table 2).

### 3.1. Adenovirus

AD contains a double-stranded DNA ranging from 26 to 45 kb, and is currently the most frequently used virus vector in cancer biotherapy. It can cause symptoms of upper respiratory tract infections [56,57]. The primary receptor for AD is the Coxsackie-adenovirus receptor (CAR), with other receptors including CD46, CD80, CD86, and desmoglein-2 (DSG2) [58]. Among oncolytic ADs, adenovirus type 5 (AD5) has been extensively studied, and can infect tumor cells through the CAR receptor [59].

In the year 2000, the National Cancer Institute in the United States initiated the phase I clinical trial of the first-generation oncolytic AD, ONYX-015, for the treatment of HNCs. This virus weakened its inhibition of the p53 gene by deleting the E1B55KD gene from its genome, thereby improving tumor targeting. In a phase II clinical trial, 37 recurrent HNSCC patients received intratumor and peritumor injections of ONYX-015. It was observed that the virus caused highly selective destruction of tumor tissue, with significant tumor regression (>50%) observed in 21% of the patients [60]. Unfortunately, the phase III clinical trial was terminated due to funding issues. OBP-301 (Telomelysin), based on AD5, was engineered by inserting the hTERT gene promoter upstream of the E1 gene. Studies have shown that combining OBP-301 with cisplatin enhances its effectiveness against HNSCC, and overcomes its resistance to radiotherapy [61]. AdGV.EGR.TNF.11D is a nonreplicating AD that expresses human TNF-α under the control of the early growth response factor 1 (EGR-1). When administered intratumorally in combination with 5-fluorouracil and hydroxyurea, it achieved an effective rate of 83.3% in recurrent HNSCC patients, with an average survival of 9.6 months [62]. KH901 is a recombinant oncolytic AD constructed through genetic engineering. It is primarily used for intratumoral injection in the treatment of recurrent HNC, and has entered phase II clinical trials [63]. In a phase I clinical trial, 23 patients received single-dose intratumoral injections of KH901 or multiple-dose injections over a period of time, and all patients showed good tolerance. The main toxicities observed were mild to moderate flu-like symptoms. No dose-limiting toxicities (DLTs) were reached in either the single-dose or multiple-dose groups, and all 23 treated patients showed an increase in AD-neutralizing antibodies. E10A is an AD that has been engineered to insert the human endostatin gene. It is primarily used for intratumoral injection in combination with paclitaxel and cisplatin for the treatment of HNSCC, and it is currently undergoing phase III clinical trials. A randomized, open-label, multicenter phase II clinical trial (NCT00634595) demonstrated that intratumoral injection of E10A in combination with paclitaxel and cisplatin could prolong progression-free survival (PFS), and improve the overall disease control rate (DCR) compared to paclitaxel and cisplatin chemotherapy alone [64]. Apart from fever, no other adverse events (AEs) were reported.

Monoclonal antibodies that target the programmed cell death protein-1/programmed cell death-ligand 1 (PD-1/PD-L1) and cytotoxic T lymphocyte-associated protein-4 (CTLA-4) pathways have brought about a permanent transformation in the treatment of various types of tumors, some of which were previously associated with a poor prognosis, including HNSCC [65,66]. The combination of OVs and ICIs in the treatment of HNC holds great research value. Several ongoing clinical trials are dedicated to exploring the combined therapeutic effectiveness of ICIs and OVs in this context [67]. VCN-01 is an oncolytic AD based on AD5, with its genome engineered for selective replication in pRB-defective tumor cells. It carries a fibroblast-specific integrin-binding motif RGD sequence for tumor targeting and expresses hyaluronidase to degrade the extracellular matrix. The efficacy and safety of VCN-01 have been confirmed in various tumor models, including HNC [68]. Clinical trials have been initiated to investigate the combination therapy of VCN-01 with durvalumab in recurrent and metastatic HNSCC (NCT03799744). AdAPT-001 is a virus derived from human AD5 that has undergone two significant modifications [69]. The first modification involves a 50 base pair deletion in the E1A promoter, which makes the virus less likely to affect normal tissues, while maintaining its ability to replicate in and destroy tumor cells. The selectivity of the virus for tumors is also linked to the impaired IFN signaling in cancer cells, which renders them more susceptible to the virus’s cytolytic effects. The second modification introduces a chimeric gene consisting of TGF-β receptor II fused with the Fc portion of human IgG-1, creating a soluble TGFβR-IgG fusion protein that effectively neutralizes the activity of the pro-oncogenic cytokine, TGF-β. As demonstrated by Christopher et al. [69], localized oncolytic infection with AdAPT-001 is not only safe, but also overcomes resistance to systemic PD-L1 immunotherapy and provides long-lasting protection against the recurrence of tumors in experiments with syngeneic tumor rechallenge. AdAPT-001 is currently under evaluation in the phase I clinical trial known as BETA PRIME, both with and without ICIs (NCT04673942). NG-641 represents an advanced adenoviral vector for tumor-specific immuno gene therapy (T-SIGn), engineered to be blood-stable and armed with transgenes [70]. The mode-of-action transgene study is a phase Ib clinical trial, conducted across multiple centers and in an open-label format, focusing on dose escalation of NG-641 as a standalone treatment or in combination with pembrolizumab (NCT04830592). Eligible patients for this study include those with newly diagnosed or recurrent locally advanced HNSCC who have definitive surgery scheduled within 8 weeks of the screening.

### 3.2. Herpes Simplex Virus

HSV is an enveloped virus containing approximately 150 kb of double-stranded DNA and encodes around 80 different proteins, which can be classified into type 1 and type 2 [71]. Due to the broad host range and the ability to carry various foreign DNA, most oncolytic HSVs entering clinical trials are modified from HSV-1 [72]. T-VEC is a recombinant HSV-1 that lacks the γ34.5 and ICP47 genes, but promotes US11 gene expression and encodes GM-CSF [73]. In preoperative lymph node injections for HNSCC, T-VEC promotes highly regressive changes in metastatic lymph nodes [74]. A phase Ib multicenter trial involving 36 patients investigated the safety and preliminary efficacy of T-VEC in combination with pembrolizumab for the treatment of platinum-resistant recurrent or metastatic HNCs (NCT02626000) [75]. The primary endpoint was DLT, and secondary endpoints included objective response rate (ORR), PFS, overall survival (OS), and safety. Most treatment-related AEs were grade 1 or 2, and treatment-related grade 2 or 3 AEs associated with T-VEC and pembrolizumab were 13.9% and 16.7%, respectively. There were no treatment-related fatal AEs. Disease control was observed in 13.9% of cases, and 10 cases (27.8%) were unable to evaluate efficacy due to early death. The median PFS and OS were 3.0 months (95% CI, 2.0–5.8 months) and 5.8 months (95% CI, 2.9–11.4 months), respectively, demonstrating the good safety profile of T-VEC.

Recent research findings indicate that T-VEC has demonstrated encouraging outcomes in the management of melanoma and sarcoma in the head and neck region. As shown in the study conducted by Franke et al. [76], the ORR for T-VEC monotherapy in cases of head and neck melanoma at the Netherlands Cancer Institute reached 80%, with half of the patients achieving a complete response (CR). The median age at the study’s outset was 78.2 years (ranging from 35 to 97), and the median follow-up period extended to 11.6 months. The data present promising outcomes and imply that T-VEC could serve as a viable alternative to systemic therapy for this specific, predominantly elderly patient group. In a phase II clinical trial reported by Kelly et al. [77], the treatment combining T-VEC and pembrolizumab exhibited antitumor activity in advanced sarcoma cases, spanning various histologic subtypes of sarcoma, while maintaining a manageable safety profile (NCT03069378). This combination therapy successfully met its predetermined primary study endpoint, and further assessments of T-VEC in conjunction with pembrolizumab for patients with specific subtypes of sarcoma are in the planning stages.

HF10 is a naturally occurring HSV with a UL56 gene deletion and has cell-fusion capability [78]. Research by Esaki et al. [79] showed that HF10 can replicate within HNSCC cells and kill them. HF10 induces tumor necrosis, CD8+ cell infiltration, and the release of antitumor cytokines, including IL-2, IL-12, TNF-α, and IFN-α, -β, -γ, to inhibit tumor growth and prolong survival. Mace et al. [80] found that HSV1716 was well-tolerated in the treatment of oral SCC, but had minimal biological activity. The main challenges include optimizing the dose, delivery, and distribution of HSV1716 into the dense heterogeneous tumor cell matrix. Increasing understanding of the interactions between HSV1716, HNSCC cells, and the immune system will help optimize antitumor efficacy. OH2, a novel oncolytic HSV-2, robustly triggers the activation of human peripheral blood mononuclear cells, resulting in heightened antitumor effectiveness in vitro and in vivo [81]. At present, it is in the initial phase of clinical trials (phase I) for the treatment of melanoma and various solid tumors (NCT03866525).

### 3.3. Other OVs

In addition to AD and HSV, various OVs have been used in clinical trials for the treatment of HNC. RV is a naturally occurring OV [82]. Oncolytics Biotech reported the data from a randomized, two-arm, double-blind, multicenter phase III clinical trial of RV in combination with standard chemotherapy for advanced stage HNC. Compared to chemotherapy alone, the combination therapy improved the median PFS of patients (94 days vs. 50 days). However, it was associated with increased side effects such as fever, chills, nausea, and diarrhea, although most patients tolerated it well [83]. Reolysin (Pelareorep) is derived from Reovirus type 3 Dearing, a naturally occurring OV that activates the RAS pathway and has cytotoxic effects on tumor cells [84,85]. The phase III clinical trial of combination therapy involving intravenous Reolysin with paclitaxel and carboplatin in HNC patients has been completed (NCT01166542). In phase I and II clinical trials [86], involving 24 patients with HNSCC and other HNCs, 1 patient achieved CR, 6 patients achieved partial response (PR), 2 patients had a major clinical response (mCR) after initial radiotherapy, 6 patients had stable disease (SD), and 5 patients experienced disease progression (DP).

MV is a negative-sense single-stranded RNA virus [87]. MV-NIS is an OV in which the sodium iodine symporter (NIS) gene is inserted into the MV genome, allowing infected cells to be imaged using single-photon emission computed tomography (SPECT) [88]. A phase I trial of intratumoral administration of MV-NIS for the treatment of HNSCC has been completed at the Mayo Clinic (NCT01846091).

VV is a double-stranded DNA virus, and most adults lack corresponding antibodies. It can infect primary and distant metastatic lesions through intravenous injection [89]. GL-ONC1 (GLV-1H68) is a VV-based oncolytic virus in which the viral thymidine kinase (TK), hemagglutinin (HA), and F14.5L genes are replaced by β-galactosidase, β-glucuronidase, and renilla luciferase/green fluorescence (RLuc-GFP), respectively [90,91]. In a phase I clinical trial, Mell et al. [91] found that intravenous injection of GL-ONC1 in combination with cisplatin chemotherapy and radiotherapy improved overall survival in late-stage HNC patients. The one-year and two-year PFS rates were 74.4% and 64.1%, and the one-year and two-year OS rates were 84.6% and 69.2%, respectively. Pexa-Vec is an oncolytic VV engineered with a deletion in the thymidine kinase gene and carries transgenes for GM-CSF and β-galactosidase [92]. A phase I clinical trial has been completed to evaluate the intratumoral administration of Pexa-Vec in combination with the CTLA-4 inhibitor ipilimumab for patients with metastatic or advanced tumors (NCT02977156).

VSV-hIFNβ-NIS is an oncolytic VSV that expresses human IFN-β and NIS, known to induce rapid and potent tumor regression with systemic treatment [93]. VSV-hIFNβ-NIS is involved in two phase I combination trials: one combines it with the anti-PD-L1 antibody avelumab for patients with refractory metastatic solid tumors (NCT02923466), and the other combines it with the anti-PD1 antibody pembrolizumab for patients with select solid tumors (NCT03647163). MEDI5395 is a recombinant Newcastle disease virus (NDV) carrying a GM-CSF transgene [94]. Recently, MEDI5395 has entered a phase I trial in combination with the PD-L1 inhibitor durvalumab (NCT03889275).

## 4. Advantages and Limitations of OVT

Compared to traditional cancer treatments, OVs rely less on specific receptor expression and are less susceptible to mutations or transcriptional resistance. They exhibit high safety and specificity, while avoiding issues of drug resistance that may arise during chemotherapy [95,96]. OVs possess multiple antitumor mechanisms and have a broad application potential, often synergizing with traditional anticancer therapies [97]. Combined therapy of standardized chemotherapy and OVs enhances the antitumor effect, ensuring safety and extending patient survival [58]. The combination of radiotherapy and OV drugs has a synergistic effect on tumor treatment [97]. Targeted drugs can increase the entry of OVs into tumor cells, synergistically enhancing their antitumor activity [98]. Combining OVT with different immunotherapies can lead to a synergistic immune response against tumors [37,99]. The use of OV in combination with various ICIs, such as CTLA-4 inhibitors and PD-L1/PD-1 inhibitors, often results in enhanced efficacy [35,67,100].

However, upon entry into the body, the host’s antiviral defense mechanisms pose a significant limitation to current OVT. This can lead to insufficient levels of OVs targeting tumors, making it challenging to achieve the desired therapeutic effect on tumors [101]. Current research suggests that carrier-based OV delivery systems may offer a potential solution to overcome this limitation. Additionally, after OVs induce a strong immune response, the body may experience adverse reactions such as fever and flu-like symptoms [102].

To comprehensively address the limitations and AEs associated with OVT, it is essential to devise a multifaceted research strategy that includes the following planning and potential solutions:

Firstly, optimizing viral virulence and safety requires advanced genetic engineering techniques to modify OVs. The aim is to reduce their virulence, ensuring safety, while still maintaining the ability to replicate and selectively lyse tumor cells. Enhancing the selectivity of OVs for tumor cells is crucial and can involve the use of tumor-specific promoters to drive viral replication, minimizing the risk of infection in healthy tissues. E1A is a crucial gene in the replication of AD and is the initial gene expressed during oncolytic adenoviral infection. To enhance the tumor-specific antitumor activity of AD, numerous tumor-specific promoters have been strategically employed to drive E1A expression [103]. These promoters include the human telomerase reverse transcriptase promoter (hTERT), the hypoxia-responsive promoter (HRE), the prostate-specific antigen promoter (PSA), the alpha-fetoprotein promoter (AFP), the alpha-lactalbumin promoter (ALA), and the mucin1 promoter (DF3/MUC1) [104,105]. However, it is important to note that this approach is applicable to only a limited number of viruses, such as AD and HSV. In the case of many other viruses, particularly RNA viruses and some DNA viruses such as VV, they operate with their own transcriptional systems, and host cell promoters are not active within the viral genome.

Secondly, improving targeting efficiency and immune evasion can be achieved through the development of immune evasion strategies to prevent OVs from being eliminated by the host immune system. Various strategic approaches have been explored to improve the transport of OVs and circumvent immune surveillance within the TME. These methods encompass the use of cytokine-induced cytotoxic cells, neural stem cells, mesenchymal stem cells, dental pulp stem cells, and irradiated tumor cells for viral delivery [106,107,108,109]. Additionally, nanoparticles, liposomes, polyethylene glycol, and polymeric particles have been harnessed to convey OVs from the systemic circulation to cancer cells [110,111,112]. Notably, synthetic nanoparticle-coated OVs exhibit extended persistence and resist viral clearance by antibodies. Furthermore, promising techniques such as ultrasound and magnetic drug targeting systems are also being investigated [113,114,115]. Cell fusion presents a strategy to facilitate the virus’s spread to adjacent cells, effectively overcoming the challenges of limited diffusion within the TME [116,117,118]. This approach involves both naturally fusogenic viruses and engineered fusogenic viruses. Naturally occurring fusogenic viruses include NDV, Sendai virus, and respiratory syncytial virus [118]. In the case of HSV-GALV, researchers employ oncolytic HSV as the foundational virus and introduce a cell-fusible fusion protein derived from the gibbon ape leukemia virus to enhance its oncolytic potential [119].

Lastly, overcoming technical challenges necessitates the development of streamlined processes for mass production of OVs. This includes optimizing viral production techniques and bioreactor systems to ensure an adequate supply of therapeutic doses. Researching innovative storage solutions and stability-enhancing techniques to maintain viral titer over time is crucial for product shelf life and ease of distribution. Implementing rigorous quality control measures to guarantee the consistency and safety of OV products involves regular testing and assessment of virus preparations to meet regulatory standards. Overall, a comprehensive research agenda that focuses on genetic modification, immune system interaction, and practical considerations such as mass production and quality control is essential. Collaborations between virologists, immunologists, genetic engineers, and pharmaceutical experts are vital to address these multifaceted challenges and advance the field of OVT. The ultimate goal is to develop safe, effective, and accessible OV treatments for a wide range of cancer patients.

## 5. Conclusions and Perspectives

Many basic and clinical studies have been conducted on OVs, but the number of OV drugs that have successfully transitioned to the market is limited and cannot meet practical demands. Future research can focus on how to enhance efficacy and expand application areas. The antitumor effects of OVs depend on the interaction between the virus, tumor cells, and the body’s immune response. Modifying OVs is a systematic endeavor that requires consideration of various factors to prepare products with clinical application prospects. The effects of making the virus express new genes through genetic modification require further exploration. In this process, a comprehensive assessment of a series of questions may be key to developing a new generation of OVs, including the selection of OV carriers, the choice of tumor types, research on the interaction between the virus and internal tumor genes, and the role of the virus in the TME.

To address the issue of less-than-ideal effectiveness of single OV treatment methods, numerous studies indicate that combining OVT with other treatment methods such as radiotherapy, chemotherapy, hyperthermia, and other immunotherapies can improve the treatment outcomes for HNC. However, more clinical data is still needed to support the effectiveness and safety of combination therapy. In the future, more high-quality clinical trials can be conducted through collaboration while continuing to develop more effective OV drugs and delivery systems, exploring ways to improve combination therapy strategies, and thereby better utilizing the potential of OVs to improve the prognosis of cancer patients.

Clinical trials present innovative therapeutic options for patients, frequently introducing novel approaches that often evolve into the subsequent standard of care. One prominent challenge in clinical trials of OVT is the inconsistency in study conditions across different trials. These conditions can vary significantly, encompassing factors such as the choice of OV, dosing regimens, administration methods, patient selection criteria, and even outcome assessment metrics. This variability makes it difficult to directly compare and synthesize findings across trials, hindering the development of standardized treatment protocols and evidence-based guidelines. For instance, a single research group conducted two significant studies on the treatment of HNSCC with T-VEC [74,75]. In the first study, they treated untreated stage III/IV patients with T-VEC, chemoradiotherapy, and cervical dissection, achieving positive outcomes. However, in the more recently published second study, they administered T-VEC in combination with pembrolizumab or pembrolizumab alone to patients with recurrent/metastatic HNSCC. This study did not show any additional benefit from T-VEC, and no further phase III trials were conducted. A more recent study reinforced this finding, as a phase III trial in advanced melanoma patients showed that the combination of T-VEC and pembrolizumab did not provide any additional clinical benefits compared to using pembrolizumab alone [120]. The implications of these findings for clinical decision-making are indeed significant. Clinicians and researchers should consider the careful selection of patients when contemplating the use of OV plus ICI therapy. Patient selection criteria should take into account factors such as tumor type, stage, genetic markers, and previous treatments. Moreover, the patient’s overall health, immune status, and prior exposure to viruses must be carefully assessed to maximize the likelihood of a positive response. Developing standardized guidelines for patient selection and stratification is essential to ensure the effectiveness of clinical trials and the broader applicability of OVT in oncology. Ongoing investigations should aim to elucidate the mechanistic insights behind the limited efficacy observed and identify strategies to enhance the synergistic potential of OVs and ICIs. Additionally, considering alternative combinations or sequencing of treatments could be explored to maximize therapeutic outcomes.

Cancer treatment is a long and systematic process, and personalized approaches such as next-generation sequencing, tumor tissue origin gene testing, neoantigen prediction, and immunological analysis can be employed to establish new treatment plans and enhance efficacy. In conclusion, in the field of targeted cancer therapy, OVs have significant clinical application potential, and it is believed that with the progression of a series of studies, more cancer patients will benefit from treatment with OV drugs.

## Figures and Tables

**Table 1 cancers-15-05291-t001:** Catalog of authorized OVs.

Virus Type	Virus Name	Modification	Year Approved	Country Approved	Primary Indication
Picornavirus	Rigvir (ECHO-7)	Unmodified	2004	Latvia	Melanoma
Adenovirus	Oncorine (H101)	Deleted for viral E1B-55K and with four deletions in viral E3	2005	China	HNC
Herpes Simplex Virus	T-VEC (Imlygic)	Deletion of ICP34.5 and ICP47, encoding two copies of human GM-CSF	2015	United States and Europe	Metastatic melanoma
	Delytact (Teserpaturev/G47Δ)	Deletion of ICP34.5, ICP6, and α47 genes	2021	Japan	Malignant glioma or any primary brain cancer

**Table 2 cancers-15-05291-t002:** Clinical trials of OVT for HNC.

Virus Type	Virus Name	Clinical Phase	Route of Administration	Cotherapy	Type of Cancer	Status	ClinicalTrials.Gov ID
Adenovirus	ONYX-015	II	i.t.	cisplatin and fluorouracil	HNSCC	withdrawn	NCT00006106
	OBP-301	II	i.t.	pembrolizumab and SBRT	HNSCC	terminated	NCT04685499
	AdGV.EGR.TNF.11D	I	i.t.	RT + 5FU + hydroxyurea	HNSCC	completed	__
	KH901	II	i.t.	__	HNC	completed	__
	E10A	III	i.t.	paclitaxel + cisplatin	HNSCC	unknown	NCT00634595
	VCN-01	I	i.t.	durvalumab	HNC	active, not recruiting	NCT03799744
	AdAPT-001	I	i.t.	ICIs	solid tumor	recruiting	NCT04673942
	NG-641	Ib	i.v.	pembrolizumab	HNSCC	recruiting	NCT04830592
Herpes Simplex Virus	T-VEC	I/II	i.t.	RT + cisplatin	HNSCC	terminated	NCT01161498
		Ib/III	i.t.	pembrolizumab	HNSCC	completed	NCT02626000
	T-VEC	II	i.t.	pembrolizumab	sarcoma	active, not recruiting	NCT03069378
	HF10	I	i.t.	__	HNSCC, breast cancer, pancreatic cancer, melanoma	completed	NCT01017185
	OH2	I	i.t.	HX 008	solid tumor, gastrointestinal cancer	recruiting	NCT03866525
Reovirus	Reolysin	III	i.v.	carboplatin, paclitaxel	solid tumor	completed	NCT01166542
Measles Virus	MV-NIS	I	i.t.	__	solid tumor	completed	NCT01846091
Vaccinia Virus	GL-ONC1	I	i.v.	RT + cisplatin	HNSCC	completed	NCT01584284
	Pexa-Vec	I	i.t.	ipilimumab	solid tumor	completed	NCT02977156
Vesicular Stomatitis Virus	VSV-IFNβ-NIS	I	i.t./i.v.	avelumab	solid tumor	completed	NCT02923466
	VSV-IFNβ-NIS	I/II	i.v.	pembrolizumab	solid tumor	recruiting	NCT03647163
Newcastle Disease Virus	MEDI5395	I	i.v.	durvalumab	solid tumor	recruiting	NCT04830592

RT, radiotherapy; i.t., intratumoral; i.v., intravenous.

## Data Availability

The data can be shared up on request.

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
