# Peer review of "Clinical Advances and Future Directions of Oncolytic Virotherapy for Head and Neck Cancer"

_cancers, 2023, doi:10.3390/cancers15215291_

Round 1

Reviewer 1 Report

Comments and Suggestions for Authors

The authors provide a comprehensive review of the development of oncolytic virotherapy and its application in the clinical treatment of head and neck cancer (HNC).

Many oncolytic viruses have been developed and are in clinical trials for various types of malignancies. Many oncolytic viruses are currently in the clinical trial phase in combination with other therapies such as radiation therapy, chemotherapy, and immunotherapy. However, not all viruses have been tested in HNC. Thus, information for clinical advances in HNC is limited. Nevertheless, this paper will contribute to the development of tumor-eluting viral therapies in HNC.

Some comments as follows.

Although adenovirus has been used clinically to treat HNC, T-vec (HSV-1) was the first oncolytic virus approved by the FDA and may be currently the primary oncolytic virus in clinical studies of oncolytic virus therapy for HNC. Recent studies have reported that T-Vec has shown promising results in the treatment of head and neck melanoma and sarcoma. Although SCC accounts for 90% of head and neck malignancies, the treatment of other malignancies of the head and neck with T-vec could also be included in this review as a notable therapeutic advance. Examples of relevant papers are listed below.
1) Franke V et al. Talimogene laherparepvec monotherapy for head and neck melanoma patients. Melanoma Res. 2023;33(1):66-70. doi: 10.1097/CMR.0000000000000866.

2) Kelly CM et al. Objective response rate among patients with locally advanced or metastatic sarcoma treated with talimogene laherparepvec in combination with pembrolizumab: A phase 2 clinical trial. JAMA Oncol. 2020;6(3):402-408. doi: 10.1001/jamaoncol.2019.6152.

Table 1: Authors should check out the following recent papers: a review article by Zheng et al. on ongoing clinical trials, including NG-350A (adenovirus), VCN-01 (adenovirus) + durvalumub, OH2 (HSV-2) + HX008. Another review article by Hwang et al. lists oncolytic viruses and immune checkpoint inhibitors. Their Table 1 includes Pexa-Vec + ipilimumab, VSV-IFNb-NIS + avelumab, MEDI5395 (NDV) + durvalumab, and others.
The two review articles above are as follows.
1) Zheng M et al. Oncolytic viruses for cancer therapy: Barriers and recent advances.
Mol Ther Oncolytics. 2019;15:234-247. doi: 10.1016/j.omto.2019.10.007.

2) Hwang JK et al. Oncolytic viruses and immune checkpoint inhibitors: Preclinical developments to clinical trials. Int J Mol Sci. 2020;21(22):8627. doi: 10.3390/ijms21228627.

Table 1: figure legends should be improved. What means by AD, RV, MV, and VV?

A problem with clinical trials is that study conditions vary from study to study. For example, two major studies by the same group have been published on the treatment of HNSCC with T-vec (Harrington et al 2010 (Ref 60), 2020 (Ref 61)). The first study treated untreated stage III/IV patients with T-vec, chemoradiotherapy, and cervical dissection with favorable results. In the second, recently published study, patients with recurrent/metastatic HNSCC received T-vec plus pembrolizumab or pembrolizumab alone; no additional benefit of T-vec was demonstrated and no further phase III trials were conducted. In the latter study, 36.1% of patients had distant metastases and 55.6% had oral SCC; T-vec was injected into cutaneous, subcutaneous, and nodal lesions, but not mucosal lesions. Therefore, it is premature to conclude that T-vec plus immune checkpoint inhibitors is not effective in the treatment of HNSCC based on the results of the latter study. The differences in the respective study designs must be fully taken into account when evaluating the efficacy of oncolytic virotherapy. Such a discussion is needed in the Limitations and/or Prospects section.

Page 4, line 166: Does Ref 23 describe phase III trial with E10A?

Author Response

The authors provide a comprehensive review of the development of oncolytic virotherapy and its application in the clinical treatment of head and neck cancer (HNC).

Reply: We are committed to continually improving our manuscript based on the insightful feedback provided by the reviewers. Based on the reviewers' feedback, we have made significant revisions to the manuscript, incorporating a wealth of new content.

Many oncolytic viruses have been developed and are in clinical trials for various types of malignancies. Many oncolytic viruses are currently in the clinical trial phase in combination with other therapies such as radiation therapy, chemotherapy, and immunotherapy. However, not all viruses have been tested in HNC. Thus, information for clinical advances in HNC is limited. Nevertheless, this paper will contribute to the development of tumor-eluting viral therapies in HNC.

Reply: We believe that this article will promote further clinical research on oncolytic virus therapy for head and neck tumors, ultimately benefiting patients.

Although adenovirus has been used clinically to treat HNC, T-vec (HSV-1) was the first oncolytic virus approved by the FDA and may be currently the primary oncolytic virus in clinical studies of oncolytic virus therapy for HNC. Recent studies have reported that T-Vec has shown promising results in the treatment of head and neck melanoma and sarcoma. Although SCC accounts for 90% of head and neck malignancies, the treatment of other malignancies of the head and neck with T-vec could also be included in this review as a notable therapeutic advance. Examples of relevant papers are listed below.

1) Franke V et al. Talimogene laherparepvec monotherapy for head and neck melanoma patients. Melanoma Res. 2023;33(1):66-70. doi: 10.1097/CMR.0000000000000866.

2) Kelly CM et al. Objective response rate among patients with locally advanced or metastatic sarcoma treated with talimogene laherparepvec in combination with pembrolizumab: A phase 2 clinical trial. JAMA Oncol. 2020;6(3):402-408. doi: 10.1001/jamaoncol.2019.6152.

Reply: We have summarized recent clinical studies of T-VEC in head and neck melanoma and sarcomas as per your suggestion, supplementing the information in section 3.2 HSV.

Table 1: Authors should check out the following recent papers: a review article by Zheng et al. on ongoing clinical trials, including NG-350A (adenovirus), VCN-01 (adenovirus) + durvalumub, OH2 (HSV-2) + HX008. Another review article by Hwang et al. lists oncolytic viruses and immune checkpoint inhibitors. Their Table 1 includes Pexa-Vec + ipilimumab, VSV-IFNb-NIS + avelumab, MEDI5395 (NDV) + durvalumab, and others.

The two review articles above are as follows.

1) Zheng M et al. Oncolytic viruses for cancer therapy: Barriers and recent advances. Mol Ther Oncolytics. 2019;15:234-247. doi: 10.1016/j.omto.2019.10.007.

2) Hwang JK et al. Oncolytic viruses and immune checkpoint inhibitors: Preclinical developments to clinical trials. Int J Mol Sci. 2020;21(22):8627. doi: 10.3390/ijms21228627.

Reply: We've removed and substituted any unsuitable citations while making corresponding adjustments to the material. We have also incorporated nine new clinical trials into the table, removed the references column, and added registration numbers and the clinical trial status. Additionally, we have introduced a new table that furnishes details regarding oncolytic virus products that have already been commercialized.

Table 1: figure legends should be improved. What means by AD, RV, MV, and VV?

Reply: We have corrected inappropriate abbreviations in the table.

A problem with clinical trials is that study conditions vary from study to study. For example, two major studies by the same group have been published on the treatment of HNSCC with T-vec (Harrington et al 2010 (Ref 60), 2020 (Ref 61)). The first study treated untreated stage III/IV patients with T-vec, chemoradiotherapy, and cervical dissection with favorable results. In the second, recently published study, patients with recurrent/metastatic HNSCC received T-vec plus pembrolizumab or pembrolizumab alone; no additional benefit of T-vec was demonstrated and no further phase III trials were conducted. In the latter study, 36.1% of patients had distant metastases and 55.6% had oral SCC; T-vec was injected into cutaneous, subcutaneous, and nodal lesions, but not mucosal lesions. Therefore, it is premature to conclude that T-vec plus immune checkpoint inhibitors is not effective in the treatment of HNSCC based on the results of the latter study. The differences in the respective study designs must be fully taken into account when evaluating the efficacy of oncolytic virotherapy. Such a discussion is needed in the Limitations and/or Prospects section.

Reply: We strongly concur with your viewpoint and have cited these cases in the Conclusion and Perspective section that supports this opinion. Additionally, we have provided our own insights and proposed strategies in response to this.

Page 4, line 166: Does Ref 23 describe phase III trial with E10A?

Reply: We have noticed that reference 23 does not describe the clinical trials of this drug. We value your expertise and would appreciate any additional insights to further enhance the quality of our manuscript.

Reviewer 2 Report

Comments and Suggestions for Authors

The authors have summarized studies of oncolytic virotherapy in H&N cancer, with focus on clinical advances. However, the current version leaves a lot to be desired.

First, this review focuses on clinical advances and future directions of oncolytic virotherapy for H&N cancer. Frankly, I did not see either one. The Table list 11 clinical trials with OVs, all completed and published. I can read many recent review articles on this topic and find all this type of  information and more. A quality review on the topic need at least another Table, listing ongoing clinical trials using OVs in HNC.

As for future directions, the authors presented three areas of research, all in 8 lines without any solutions and strategies (lines 257-264). The authors need to greatly expand the discussion and come up with some solid planning, strategies and potential solutions.

Second, the whole manuscript lacked discussion of basic research to provide readers fundamental information on oncolytic virotherapy. In lines 68-77, this paragraph is the only paragraph describing mechanisms of action by OVs. The authors mentioned that OVs could transform cold tumors hot and induce immunogenic cell death. There are two major issues. (1). The summary is too simple, too short. (2). There is only one reference cited (ref #27). (a). One reference is not enough. (b). That reference, unfortunately, is a useless one as it is not published in a peer-review journal, and most readers (and reviewers included) do not have access to it. We need high quality, peer-reviewed papers to support such important conclusions and statements. There are plenty of such articles in the literature. Please include a few to support your statements.

In summary, this is a poorly written review. It needs very extensive revision before it is suitable for this quality journal.

Some minor issues are,

1.      Lines 61-67. The authors discussed 1-3rd generations of OVs. It seems that there are fourth generation of OVs now. Please see [PMID: 37491263] Wang X et al., Oncolytic virotherapy evolved into the fourth generation as tumor immunotherapy. Journal of Translational Medicine, 2023, 21:500.

2.      The current references:

(1). Format: It is not consistent from one to another. This reviewer would suggest that authors delete the “month date” from the listing. In addition, when listing author names, you should use “et al.” when there are too many authors but cannot use “…” as the authors have done in ref #34.

(2). Ref #38. This is a paper published online in 2020. The article number (replacing page numbers) is still missing. How could that happen? The PubMed database did a poor job without updating the information. By going to the journal website, I found that the article number is e2020047.

Author Response

The authors have summarized studies of oncolytic virotherapy in H&N cancer, with focus on clinical advances. However, the current version leaves a lot to be desired.

Reply: We are committed to continually improving our manuscript based on the insightful feedback provided by the reviewers.

First, this review focuses on clinical advances and future directions of oncolytic virotherapy for H&N cancer. Frankly, I did not see either one. The Table list 11 clinical trials with OVs, all completed and published. I can read many recent review articles on this topic and find all this type of information and more. A quality review on the topic need at least another Table, listing ongoing clinical trials using OVs in HNC.

Reply: In the "Advances of OV Treatment for HNCs" section, we have introduced six new viruses for the treatment of head and neck cancers, including AdAPT-001. We have provided information on the combination of viruses with immune checkpoint inhibitors and presented clinical research results on T-VEC's efficacy in treating head and neck melanoma and sarcomas. In the "Conclusion and Perspectives" section, we have included a new paragraph that focuses on the challenges associated with clinical trials. We have incorporated nine new clinical trials into the table, removed the references column, and added registration numbers and the clinical trial status. Additionally, we have introduced a new table providing information on oncolytic virus products that are already on the market.

As for future directions, the authors presented three areas of research, all in 8 lines without any solutions and strategies (lines 257-264). The authors need to greatly expand the discussion and come up with some solid planning, strategies and potential solutions. Second, the whole manuscript lacked discussion of basic research to provide readers fundamental information on oncolytic virotherapy. In lines 68-77, this paragraph is the only paragraph describing mechanisms of action by OVs. The authors mentioned that OVs could transform cold tumors hot and induce immunogenic cell death. There are two major issues. (1). The summary is too simple, too short. (2). There is only one reference cited (ref #27). (a). One reference is not enough. (b). That reference, unfortunately, is a useless one as it is not published in a peer-review journal, and most readers (and reviewers included) do not have access to it. We need high quality, peer-reviewed papers to support such important conclusions and statements. There are plenty of such articles in the literature. Please include a few to support your statements.

Reply: We have substantially broadened the discussion and formulated robust plans, strategies, and prospective solutions for future directions. We have replaced this reference with "Oncolytic Viruses: Newest Frontier for Cancer Immunotherapy" from the Cancers and supplemented the specific mechanisms.

In summary, this is a poorly written review. It needs very extensive revision before it is suitable for this quality journal.

Reply: Cancers is an international peer-reviewed journal that focuses on both basic research and clinical applications in the field of cancer. We have noticed that in recent years, this journal has published numerous articles on clinical advancements in the treatment of H&N cancer using various approaches. Therefore, Cancers is our first choice for submitting this manuscript.

Some minor issues are,

  1.      Lines 61-67. The authors discussed 1-3rd generations of OVs. It seems that there are fourth generation of OVs now. Please see [PMID: 37491263] Wang X et al., Oncolytic virotherapy evolved into the fourth generation as tumor immunotherapy. Journal of Translational Medicine, 2023, 21:500.
  2.      The current references:

(1). Format: It is not consistent from one to another. This reviewer would suggest that authors delete the “month date” from the listing. In addition, when listing author names, you should use “et al.” when there are too many authors but cannot use “…” as the authors have done in ref #34.

 (2). Ref #38. This is a paper published online in 2020. The article number (replacing page numbers) is still missing. How could that happen? The PubMed database did a poor job without updating the information. By going to the journal website, I found that the article number is e2020047.

Reply: Oncolytic viruses can be classified into three generations based on their development. The references you provided assert that oncolytic virus therapy represents a fourth-generation immunotherapy, distinct from surgery, radiation, and chemotherapy. We have reviewed and rectified issues related to reference formatting and specific details. We value your expertise and would appreciate any additional insights to further enhance the quality of our manuscript.

Round 2

Reviewer 2 Report

Comments and Suggestions for Authors

During the revision, the authors have made some significant improvements. However, addition of some details and minor corrections are needed.

Some minor issues are,

1.      Under Section 4. The authors stated that “Enhancing the selectivity of OVs for tumor cells is crucial and can involve the use of tumor-specific promoters to drive viral replication, minimizing the risk of infection in healthy tissues.” (Lines 326-328). The statements need further refinement, as it is applicable only to a few viruses such as Adenovirus and HSV. Many other viruses, including most if not all of the RNA viruses, and some DNA viruses (e.g., vaccinia virus), you cannot use tumor-specific promoters that come from mammalian genomes. These virus use their own transcriptional system and the host cell promoters are not active in the viral genome.

2.      Lines 332-340. “The use of immune evasion to prevent OVs from being eliminated by the host immune system” is indeed a strategy and has been studied in multiple papers. For example, [PMID: 20703311] The combination of immunosuppression and carrier cells significantly enhances the efficacy of oncolytic poxvirus in the pre-immunized host. Gene Ther. 2010 Dec;17(12):1465-75.

The authors may expand the discussion and include the use of carrier cells as another strategy for immune evasion (cite this and a few other references).

3.      Line 365. TME for tumor microenvironment. As in the whole manuscript, this is the only place to use tumor microenvironment, then the abbreviation is not needed. In addition, I am a bit surprised that the phrase “tumor microenvironment” has not been used  for a few times in such a review article.

4.      Lines 382-384. “For instance, a single research group conducted two significant studies on the treatment of HNSCC with T-VEC (Harrington et al. 2010 (Ref 60) and 2020 (Ref 61)).” Please cite the reference in the right format.

5.      Lines 375-399. In this paragraph, authors have discussed the results of two clinical trials, one of them was the use of “T-VEC in combination with pembrolizumab or pembrolizumab alone to patients with recurrent/metastatic HNSCC. This study did not show any additional benefit from T-VEC, and ....”

Interestingly, another study further supported this observation. A phase III trial of T-VEC and Pembrolizumab in advanced melanoma patients failed to achieve additional clinical benefit than Pembrolizumab alone.

[PMID: 35998300] Chesney JA, Ribas A, Long GV, Kirkwood JM, Dummer R, Puzanov I, Hoeller C, Gajewski TF, Gutzmer R, Rutkowski P, et al. Randomized, Double-Blind, Placebo-Controlled, Global Phase III Trial of Talimogene Laherparepvec Combined With Pembrolizumab for Advanced Melanoma. J Clin Oncol. 2023;41:528-540.

Please cite this paper and discuss it in the context.

6.      One abbreviation can be used for only one thing. However, the authors have used OV for either oncolytic virus (defined on lines 21 and 25 in the Abstract) and then for oncolytic virotherapy (lines 321-322). It is suggested that authors use OVT for oncolytic virotherapy (as used by others). 

Comments on the Quality of English Language

Minor changes are needed.

Author Response

  1.      Under Section 4. The authors stated that “Enhancing the selectivity of OVs for tumor cells is crucial and can involve the use of tumor-specific promoters to drive viral replication, minimizing the risk of infection in healthy tissues.” (Lines 326-328). The statements need further refinement, as it is applicable only to a few viruses such as Adenovirus and HSV. Many other viruses, including most if not all of the RNA viruses, and some DNA viruses (e.g., vaccinia virus), you cannot use tumor-specific promoters that come from mammalian genomes. These virus use their own transcriptional system and the host cell promoters are not active in the viral genome.
  2.      Lines 332-340. “The use of immune evasion to prevent OVs from being eliminated by the host immune system” is indeed a strategy and has been studied in multiple papers. For example, [PMID: 20703311] The combination of immunosuppression and carrier cells significantly enhances the efficacy of oncolytic poxvirus in the pre-immunized host. Gene Ther. 2010 Dec;17(12):1465-75.

The authors may expand the discussion and include the use of carrier cells as another strategy for immune evasion (cite this and a few other references).

Reply: We highly appreciate your perspective, and we have discussed and supplemented it in the manuscript.

  1.      Line 365. TME for tumor microenvironment. As in the whole manuscript, this is the only place to use tumor microenvironment, then the abbreviation is not needed. In addition, I am a bit surprised that the phrase “tumor microenvironment” has not been used  for a few times in such a review article.

Reply: We recognize the significance of the "tumor microenvironment," and we have increased the use of this term in this revision.

  1.      Lines 382-384. “For instance, a single research group conducted two significant studies on the treatment of HNSCC with T-VEC (Harrington et al. 2010 (Ref 60) and 2020 (Ref 61)).” Please cite the reference in the right format.

Reply: We have addressed this issue.

  1.      Lines 375-399. In this paragraph, authors have discussed the results of two clinical trials, one of them was the use of “T-VEC in combination with pembrolizumab or pembrolizumab alone to patients with recurrent/metastatic HNSCC. This study did not show any additional benefit from T-VEC, and ....”

Recently, another study further supported this observation. A phase III trial of T-VEC and Pembrolizumab in advanced melanoma patients failed to achieve additional clinical benefit than Pembrolizumab alone. The implications of these findings for clinical decision-making are indeed significant. Clinicians and researchers should consider the careful selection of patients and cancer types when contemplating the use of OV plus ICIs therapy. While it may not universally apply, there may be certain subpopulations or cancer types where this combination therapy still holds promise. Ongoing investigations should aim to elucidate the mechanistic insights behind the limited efficacy observed and identify strategies to enhance the synergistic potential of OV and ICIs. Additionally, considering alternative combinations or sequencing of treatments could be explored to maximize therapeutic outcomes.

[PMID: 35998300] Chesney JA, Ribas A, Long GV, Kirkwood JM, Dummer R, Puzanov I, Hoeller C, Gajewski TF, Gutzmer R, Rutkowski P, et al. Randomized, Double-Blind, Placebo-Controlled, Global Phase III Trial of Talimogene Laherparepvec Combined With Pembrolizumab for Advanced Melanoma. J Clin Oncol. 2023;41:528-540.

Please cite this paper and discuss it in the context.

Reply: We appreciate this valuable insight, and it will certainly enhance the depth of our discussion on this crucial topic in our review. We have cited this article and presented our viewpoint.

  1. One abbreviation can be used for only one thing. However, the authors have used OV for either oncolytic virus (defined on lines 21 and 25 in the Abstract) and then for oncolytic virotherapy (lines 321-322). It is suggested that authors use OVT for oncolytic virotherapy (as used by others).

Reply: We have incorporated your suggestion and corrected this abbreviation. In addition to the changes mentioned above, we have also added some references in paragraph 2 and 4.